# Massively parallel fabrication of crack-defined gold break junctions featuring sub-3 nm gaps for molecular devices

Valentin Dubois[1], Shyamprasad N. Raja [1], Pascal Gehring [2], Sabina Caneva[2], Herre S.J. van der Zant[2], Frank Niklaus[1] & Göran Stemme[1]

Break junctions provide tip-shaped contact electrodes that are fundamental components of nano and molecular electronics. However, the fabrication of break junctions remains notoriously time-consuming and difficult to parallelize. Here we demonstrate true parallel fabrication of gold break junctions featuring sub-3 nm gaps on the wafer-scale, by relying on a novel self-breaking mechanism based on controlled crack formation in notched bridge structures. We achieve fabrication densities as high as 7 million junctions per cm$^2$, with fabrication yields of around 7% for obtaining crack-defined break junctions with sub-3 nm gaps of fixed gap width that exhibit electron tunneling. We also form molecular junctions using dithiol-terminated oligo(phenylene ethynylene) (OPE3) to demonstrate the feasibility of our approach for electrical probing of molecules down to liquid helium temperatures. Our technology opens a whole new range of experimental opportunities for nano and molecular electronics applications, by enabling very large-scale fabrication of solid-state break junctions.

[1] Department of Micro and Nanosystems (MST), School of Electrical Engineering and Computer Science (EECS), KTH Royal Institute of Technology, SE-10044 Stockholm, Sweden. [2] Kavli Institute of Nanoscience, Delft University of Technology, Lorentzweg 1, 2628 CJ Delft, The Netherlands. These authors contributed equally: Shyamprasad N Raja, Pascal Gehring. Correspondence and requests for materials should be addressed to F.N. (email: frank.niklaus@eecs.kth.se) or to G.S. (email: goran.stemme@eecs.kth.se)

Practical molecular electronics based on solid-state devices will require the integration of arrays of interconnected molecular junctions into circuits and systems[1–5]. Before this becomes possible, new methodologies have to be developed for scalable and reproducible fabrication of nanogap electrodes featuring sub-3 nm wide gaps[6–10]. Mechanically controllable break junctions (MCBJs)[4,6,8,9,11,12], scanning tunneling microscopy based break junctions (STM-BJ)[13,14] and electromigration breakdown junctions (EBJs)[15,16] made of gold are currently the most widespread nanogap electrodes used to realize molecular junctions. For fundamental investigation of molecular junctions, reconfigurable nanogaps with sub-nm precision can be achieved using MCBJs and STM-BJs. However, MCBJs, STM-BJs, and EBJs are unsuitable for producing densely integrated individually addressable arrays of junctions due to the need for an external apparatus (motorized bending stage, piezoelectric actuators, or current source, respectively) with an electrical feedback mechanism to drive and monitor the breaking process of each metal constriction separately. Although parallel fabrication of a very limited number of break junctions (<16) has been demonstrated through electromigration[17–19], there is currently no truly parallel fabrication scheme available that can simultaneously induce breaking of thousands of metal constrictions with sufficient process control to consistently form sub-3 nm wide gaps. Previous attempts at self-aligned fabrication of greater numbers of nanogaps have exploited the lateral expansion due to oxidation of easily oxidized metals such as chromium or aluminum, and used them as sacrificial layers to create nanogaps or long nanoscale slits between two electrodes, which were defined by successive cycles of electron beam lithography and metal deposition[20–22]. These techniques are however unsuitable for creating nanogaps between atomic-scale electrode tips using a parallelizable process. The large-scale fabrication of break junctions in the range of $10^9$ junctions per chip has so far been considered as inaccessible[5,23]. Recently, a novel approach using controlled crack formation in electrode-bridge structures made of a brittle material has been proposed[24] and demonstrated[25,26] for highly parallel fabrication of sub-3 nm nanogap electrodes made of brittle electrode materials such as titanium nitride (TiN). However, this approach is not suitable for realizing nanogap electrodes made of ductile metals such as gold. Gold as electrode material is favored in many applications due to its chemical inertness and ability to covalently attach molecules in a subsequent back-end process.

Here, we introduce a new, fully scalable type of break junction, which we call crack-defined break junction (CDBJ). The methodology to realize CDBJs combines conventional wafer-scale semiconductor fabrication for the fabrication of metal constrictions, and crack formation for the highly parallel and self-induced breaking of the metal constrictions. This unique association of patterning techniques leads to a truly parallel fabrication scheme, whereby the processing time is independent of the device density on the substrate. In this study we present the fabrication of millions of crack-defined gold break junctions with sub-3 nm gaps on a wafer scale using this methodology, achieving yields of around 7% for the best design parameters. Compared to electromigrated break junctions, this is a $10^5$-fold improvement in fabrication-throughput at comparable fabrication yield. We also demonstrate the suitability of our CDBJs for studying electrical transport properties of molecules from room temperature down to 7 K by measuring molecular junctions formed by contacting oligo (phenylene ethynylene) (OPE3) using CDBJs.

## Results

**Crack-defined break junctions**. The methodology to realize the CDBJs is illustrated in Fig. 1. First, a layer of a brittle material (here titanium nitride, TiN) is deposited on a substrate that is pre-coated with a sacrificial layer (here amorphous silicon, a-Si). The TiN is deposited using process conditions that induce tensile stress at room temperature. Next, a thin layer of electrode material (here gold) is deposited on top of the TiN. Thereafter, the layer stack is patterned to outline a notched bridge (Fig. 1a). The gold-coated TiN bridge structure is then released from the substrate by undercut etching of the a-Si sacrificial layer (Fig. 1b). During this step, the internal stress in the TiN bridge structure redistributes and concentrates at the notched constriction, inducing a crack in the brittle TiN. Upon fracture, the resulting TiN cantilever pair acts as a nano-pulling stage, whereby the cantilevers spontaneously retract in opposite directions, pulling apart the section of the gold located above the crack-line (Fig. 1c). The straining of the ductile gold electrode material depends on the displacement $w$ of the cracked extremities of the TiN cantilevers, which is determined by the length $L$ of the bridge[25,26] and the elastic strain $\varepsilon$ of the layer of brittle material, with:

$$w = \varepsilon \times L. \qquad (1)$$

For the deposited brittle TiN in our experiments, $\varepsilon$ was found to be 2.7 nm μm$^{-1}$ (see Supplementary Fig. 1), indicating that,

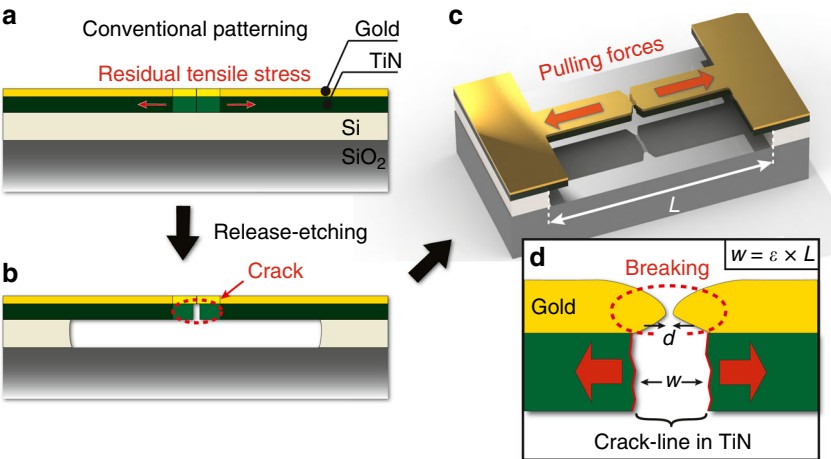

**Fig. 1** Schematics of the proposed methodology to form crack-defined break junctions. **a** A pre-stressed notched titanium nitride (TiN) bridge structure coated with a thin layer of gold is patterned; **b** the release-etching of the bridge structure induces the formation of a crack in TiN. **c** The formed TiN cantilevers retract and pull apart the section of gold located above the crack-line. **d** The pulling action $w$, defined by the length $L$ of the bridge, causes necking of the ductile gold; for a sufficiently large $w$, the gold breaks, thereby forming a nanogap with an inter-electrode separation $d$

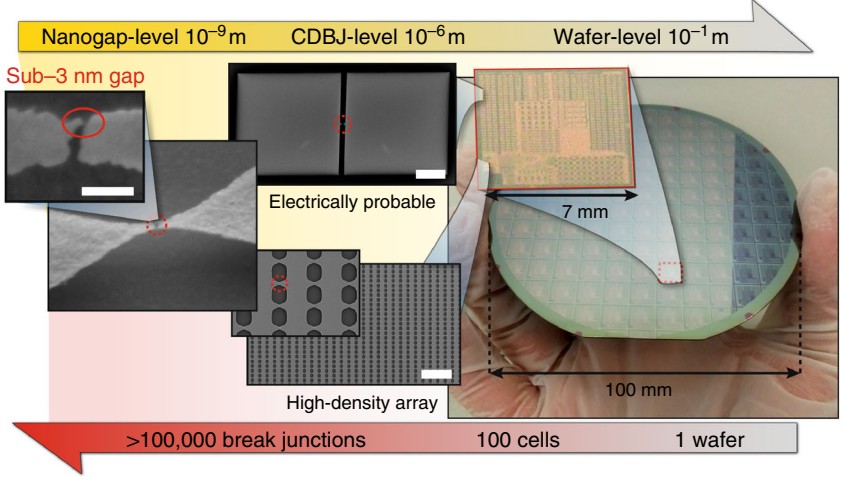

**Fig. 2** Hierarchical depiction of the fabrication of crack-defined break junctions on a wafer scale. Optical and SEM images of fabricated crack-defined break junctions (CDBJ) depicted in a range from the nano-scale (sub-3 nm gaps) to the macro-scale (100 mm diameter wafer). A total of 780,000 bridge structures were fabricated and released simultaneously using only conventional wafer-scale processes on the 100 mm diameter wafer. Upon cracking, each bridge subsequently exerted a defined pulling action on the section of gold located above the crack-line, thereby forming hundreds of thousands of CDBJs in a massively parallel fashion on wafer-scale. Scale bar is 30 μm for the "electrically probable" CDBJ, 20 μm for the "high-density array" of CDBJs, and 50 nm for the CDBJ featuring a sub-3 nm gap

after the fracture of the bridge, the TiN cantilever pair retract by a total of 2.7 nm for every micrometer of length $L$ of the bridge. For a sufficiently large displacement $w$ of a cantilever pair, the gold electrode material is pulled until breaking, thereby forming a pair of gold electrodes with an inter-electrode separation $d$ (Fig. 1d).

We fabricated CDBJs by patterning gold-coated TiN bridges on a 100 mm diameter wafer and forming break junctions in a massively parallel fashion (see Fig. 2). Adhesion of the gold layer to the underlying TiN was ensured using a 3 nm thick Cr adhesion layer. The patterning of the gold-coated bridges was done using an I-line (365 nm) stepper (resolution: ~500 nm). Following the lithography (see Methods and Supplementary Fig. 2 for details on the fabrication), about 100 identical unit cells with dimensions of $7 \times 7$ mm$^2$ were formed on the wafer, where each cell contained around 7800 bridge structures. Thus, a total of approximately 780,000 gold-coated bridge structures were produced on the wafer (see Fig. 2). All 780,000 prefabricated bridge structures were subsequently release-etched simultaneously by isotropic plasma etching of the a-Si sacrificial layer. In this step, more than 95% of the 780,000 bridges distributed across the wafer successfully cracked. Upon cracking, the formed TiN cantilevers instantaneously exerted their pre-defined pulling action on the sections of the gold layer located above the cracks. A scanning electron microscope (SEM) image of a resulting representative CDBJ featuring a sub-3 nm gap is shown in Fig. 2.

First, we investigated the yield of TiN cracking in gold-coated TiN bridges, in a high-density array using SEM. The middle SEM image in Fig. 2 shows a portion of an array of $50 \times 50 = 2500$ junctions (determined by a bridge length of $L = 2.5$ μm). Out of 1250 examined, only three bridges were found to be uncracked, thus leading to a yield of cracking for these devices higher than 99.7% for a density of 7 million junctions per cm$^2$ (see Supplementary Fig. 1 for an SEM image of the full array). Next, we investigated the yield for obtaining gold break junctions with sub-3 nm gaps on top of successfully cracked TiN bridges. This was done using a different array of bridges which could be electrically probed one at a time. Out of a total 270 probed bridges spread across the wafer, about 7% exhibited tunneling behavior, thereby indicating that sub-3 nm gaps were achieved for these devices (see Supplementary Table 1). This

demonstrates that gold break junctions featuring sub-3 nm gaps can be realized at wafer-scale with densities on the order of 490,000 devices per cm$^2$.

**Characterization of the process window of our CDBJs.** To investigate the process window of our CDBJs, different bridges were designed, each with a well-defined pulling action $w$, ranging from about 3 nm for the shortest bridges up to several hundreds of nanometers for the longest bridges. The result of the pulling actions on the gold was examined using SEM imaging, as shown in Fig. 3. At very small pulling actions $w$ of below 3 nm (determined by a bridge length of $L < 1$ μm), the 10 nm thick gold necked, but remained fully intact (Fig. 3a, b; type-1 junctions). At larger pulling actions $w$ of between 3 and 16 nm (determined by bridge lengths of $L = 1$–6 μm), the strained gold contained nanometric voids of different sizes scattered in the direction of the crack line in TiN (Fig. 3a, c; type-2 junctions). Since the voids were not sufficiently large to cause complete breakage, gold ligaments formed along the crack-line and linked the pair of TiN cantilevers (see Fig. 3g). Bridges designed with narrower notched constrictions contained as few as one or two ligaments (see Fig. 3h). At even larger pulling actions $w$, above 16 nm (determined by bridge lengths of $L > 6$ μm), breaking of the gold ligaments was accomplished and resulted in pairs of gold electrodes (Fig. 3a, d; type-3 junctions). This fracture behavior of gold at the nanoscale is consistent with the nucleation, growth, and coalescence of voids in ductile metals[27].

While the pulling action $w$ caused by the retraction of the cracked cantilevers was accurately reproduced, the breaking of the gold was stochastic and responsible for device-to-device variability. The strained gold sections at each crack-line are subjected to a combination of tensile and shear stresses that depend on the local orientation of the crack in the poly crystalline TiN[25]. Therefore, even identically designed bridges inevitably featured different distributions of ligaments across the crack-line, with variations in number, position, spacing, orientation, and shape of the ligaments. Because of this, we found a gradual transition starting at $w = 3$ nm, for which bridges yielded only type-1

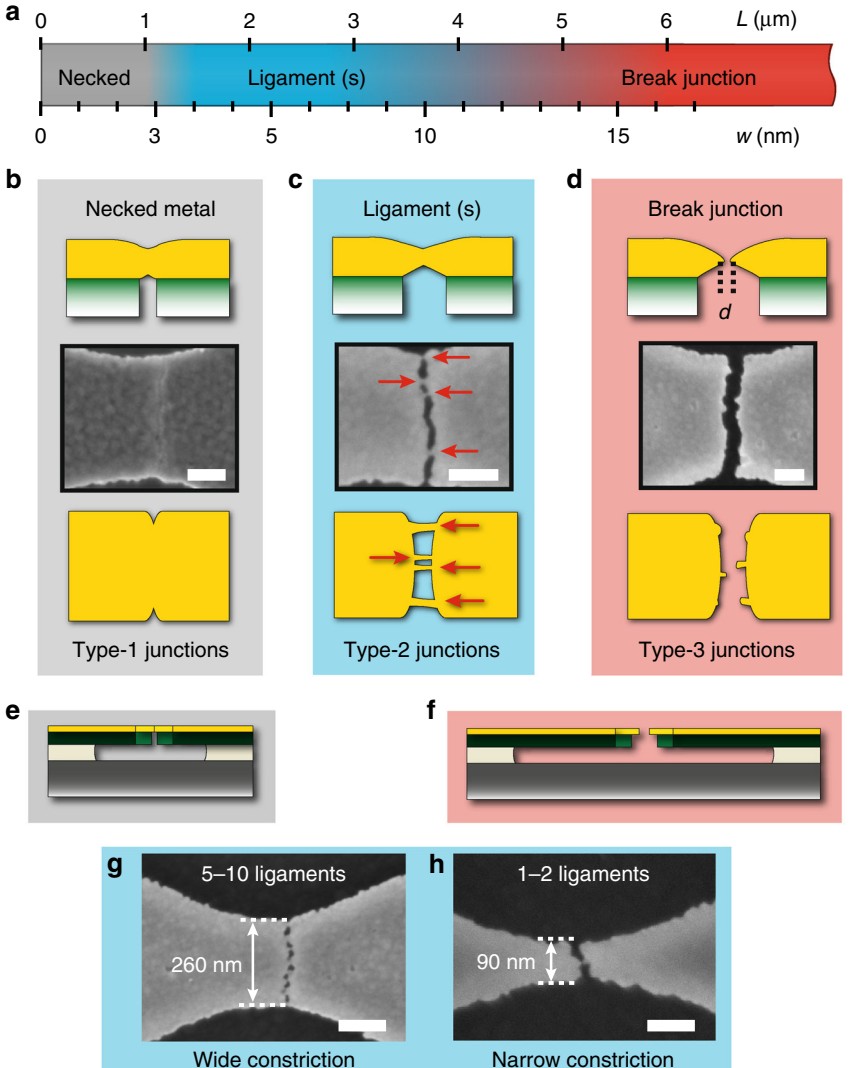

**Fig. 3** Results of the pulling action on 10 nm thick gold. **a** Each of the 780,000 bridges that were fabricated on the silicon wafer was designed to achieve a specific pulling action $w$, which was defined by the length $L$ of each individual bridge (see Eq. 1). The various pulling actions $w$ resulted in three distinct outcomes for the strained sections of gold above each crack-line. **b** When $w < 3$ nm, the 10 nm thick gold only undergoes necking (type-1 junctions); SEM imaging on these junctions reveal a lighter area where the underlying crack in the TiN has propagated. **c** When 3 nm $< w <$ 16 nm, nanometric voids appear in the gold, forming ligaments (indicated by red arrows) crossing the nanogap and connecting the electrodes (type-2 junctions). **d** When $w > 16$ nm, the gold ligaments break, thereby forming two electrodes separated by a distance $d$ (type-3 junctions). **e**, **f** Cross-section schematics depicting that short bridges and consequently small $w$ produces necked metal (**e**), whereas long bridges and large $w$ produce break junctions (**f**). **g**, **h** Two SEM images of junctions described in **c** illustrate that the average number of gold ligaments depends on the width of the constriction of the bridge. Specifically, the narrow break junction design in **h** results in 1 or 2 ligaments, and was selected for detailed electrical characterization (see Fig. 4) for its potential suitability for contacting and probing molecules. Scale bars: 100 nm

junctions (see Fig. 3a, e), to $w = 16$ nm, for which bridges yielded only type-3 junctions (see Fig. 3a, f). Bridges designed for 6 nm $< w <$ 16 nm pulling actions yielded both type-2 and type-3 junctions, with different proportions of each junction type, depending on the selected $w$. Moreover, we found that the width of the constrictions played an important role in determining the value of $w$ for which the transition from type-2 to type-3 occurred. For narrow constrictions near the average inter-ligament spacing, type-3 junctions appeared for pulling actions as small as $w = 6$ nm.

To produce nanogap electrodes suitable for electron tunneling experiments, CDBJs should ideally each form a single ligament that undergoes breaking but results in an inter-electrode separation $d$ smaller than 3 nm[3]. Assuming a normally

distributed stochastic breaking of the CDBJs, a bridge design resulting in equal numbers of type-2 and type-3 junctions would give the highest yield for 1–5 nm nanogap electrodes. In our experiments, the most promising bridge design had $w = 9$ nm (determined by a bridge length of $L = 3.3$ µm) and a 90 nm wide constriction, and produced one or two gold ligaments (see Supplementary Fig. 3 for details on this junction design). Since direct visualization of sub-3 nm gaps is beyond the resolving ability of SEM, we characterized the nanogaps electrically, by applying bias voltages and measuring the resulting currents. Thirty junctions featuring the same bridge design were probed in each cell. We characterized all 90 junctions that were present on three adjacent cells on the wafer. Out of these 90 junctions, four were discarded due to technical faults during the electrical

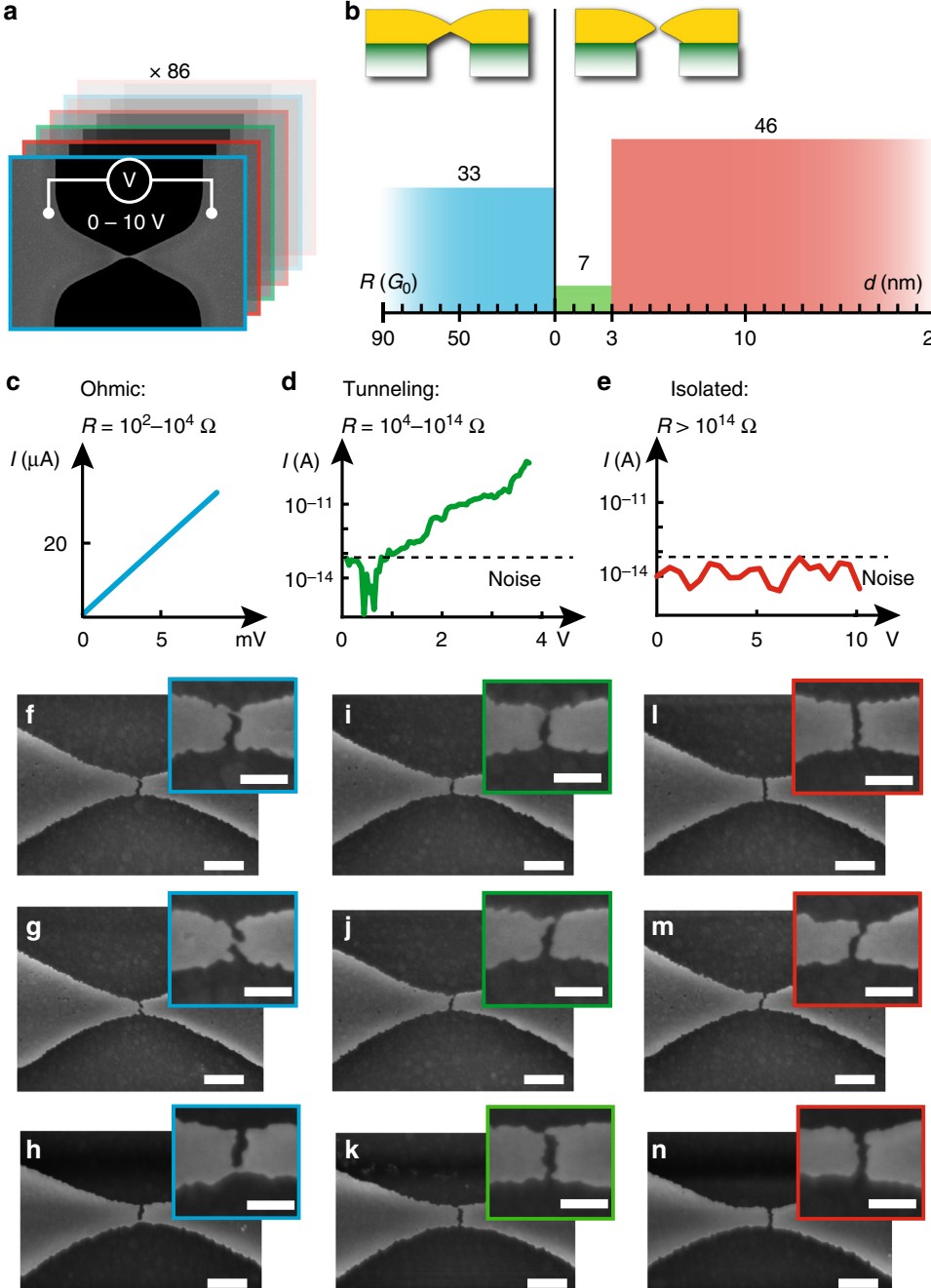

**Fig. 4** Classifying junctions using electrical characterization and SEM imaging. **a** Schematic of two probe electrical characterization, and **b** the collected results of the electrical characterization of 86 nominally identical junctions featuring an initial bridge length of $L = 3.3\,\mu m$ and a 90 nm wide constriction. Due to the stochastic breaking of the strained gold at the crack-lines of each of the 86 junctions, the resulting gold electrodes were either connected by one or two unbroken ligaments of total resistance $R$, or separated by gaps of widths $d$. Of the 86 junctions, 33 junctions showed ohmic behavior with at least one connected ligament, 46 junctions showed electrically isolated electrodes which could also be confirmed by visual inspection, and seven junctions showed tunneling $I$–$V$ characteristics thereby revealing gold ligaments that have broken and formed a sub-3 nm gap. **c**–**e** Representative $I$–$V$ characteristics of the three outcomes: ohmic, tunneling and isolated. **f**–**n** SEM images of three representative probed junctions for each of the three outcomes: ohmic (**f**–**h**), tunneling (**i**–**k**), and isolated (**l**–**n**) are shown. These SEM images illustrate the strong correlation found between the electrical characterization performed 'blind', without prior visual inspection of the junctions, and the morphology of the junctions revealed by SEM imaging. Scale bar is 200 nm for **f**–**n** and 100 nm for insets of **f**–**n**

probing or because they showed signs of contamination upon visual inspection in an optical microscope. A schematic of the electrical characterization procedure is shown in Fig. 4a and the results are summarized in Fig. 4b. The electrical characterization revealed that among the remaining 86 junctions, 33 featured

ohmic characteristics with resistances ranging from 150 Ω to 1.1 kΩ, or equivalently from 86 to 13 times the conductance quantum $G_0$ (see Fig. 4c). Further, 46 junctions did not exhibit any detectable current for applied bias up to 10 V (see Fig. 4e). Electron tunneling characteristics were observed in seven

junctions (see Fig. 4d), thereby demonstrating a yield of ~8% for sub-3 nm junctions. The gap widths ranged from 0.8 to 1.5 nm and were determined by fitting the $I$–$V$ characteristics to a one-dimensional (1-D) transmission model across a symmetric potential barrier[28]. Details of the model and fitting procedure are given in the Methods, and the fit parameters are provided in Supplementary Table 1.

After electrical characterization, each junction was visually inspected by SEM imaging to correlate the outcome of the electrical characterization with the morphology of the junctions (see Fig. 4f–n for SEM images of 9 representative junctions). In contrast to EBJs, CDBJs are suspended above the substrate surface at a distance of 200 nm. This makes it possible to obtain sharp, high-resolution images of our junctions with good contrast between the gold electrodes and the background of the gap. For each junction, we found a strong correlation between the results of the electrical and morphological characterization. As expected, junctions that had ohmic properties exhibited at least one unbroken gold ligament. The lengths and widths of these gold ligaments were consistently smaller than 10 nm. For all junctions that showed complete electrical isolation, gaps separating the gold electrodes could be clearly identified in SEM images, thereby providing visual evidence that tunneling currents could not be measured for these junctions. In some cases, the gaps appeared as small as 5 nm, at the resolution limit of the SEM. Finally, all seven junctions that showed tunneling behavior presented one ligament featuring a local narrowing, or pinching, at one extremity without distinct signs of either a gap or a continuous ligament. This visual uncertainty is also expected from sub-3 nm gaps that are below the resolution limit of the SEM. Representative SEM images of junctions with their respective $I$–$V$ characteristics are shown in Fig. 4 and a similar data set for all 30 junctions in one of the three probed cells are presented in Supplementary Note 1.

To demonstrate the potential to realize tunneling break junctions across larger wafer areas, we further inspected CDBJs in six cells positioned along the edge of the wafer (see Supplementary Fig. 4 for details on the location of these cells on the wafer). In these cells, the previous bridge design that resulted in type-2 and type-3 junctions for cells located towards the center of the wafer (see Supplementary Fig. 3) was deemed unsuitable for forming tunneling junctions, as it resulted in mainly type-3 junctions with gaps clearly distinguishable in the SEM. This was due to higher etch-rates at the wafer edge in the plasma etching processes used which caused the constrictions of the bridges to be narrower and the undercuts to be deeper. This is an indication that only small deviations in bridge geometry can be tolerated for achieving a repeatable breaking process. For these cells at the wafer edge, a different bridge design, resulting in effectively shorter and wider bridges ($L = 3$ μm and constrictions of 100 nm; see Supplementary Fig. 4 for details on this junction design), exhibited locally pinched ligaments, which is typical for tunneling junctions formed by this method. Among the 180 electrically probable junctions inspected in these cells, 31 were first identified as potential tunneling junctions using SEM. Subsequent electrical characterization revealed that 15 out of these 31 junctions exhibited measurable tunneling currents. However, since the currents only appeared at bias values exceeding 3 V in five of these junctions, we were only able to estimate gap widths in 10 of the 15 junctions using fits to the 1-D transport model. Thus, for the selected bridge design, the yield of sub-3 nm gap-widths was ~6%. The other 16 of the selected junctions emerged as connected ligaments with resistances equivalent to about 30 $G_0$. These results further demonstrate that tunneling junctions can readily be spotted via SEM imaging with an accuracy of about

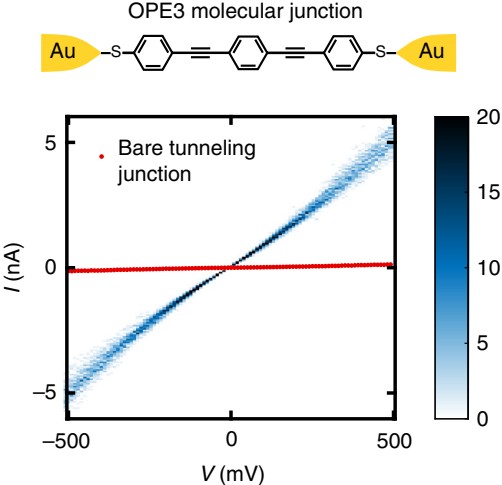

**Fig. 5** Demonstration of the formation of a molecular junction using a crack-defined break junction. The schematic is an idealized depiction of a dithiol-terminated oligo(phenylene ethynylene) molecule (OPE3) molecule deposited in a crack-defined break junction. The $I$–$V$ characteristics of tunneling gaps were measured before and after the process of OPE3 deposition and the successful formation of a molecular junction is indicated by a large increase in conductance of a tunneling gap, typically by an order of magnitude or more. The $I$–$V$ histogram of the OPE3 junction plotted here consists of 64 successive $I$–$V$ traces, and the conductance increased from $2 \times 10^{-6}$ $G_0$ to $1 \times 10^{-4}$ $G_0$ after OPE3 deposition

50%. To the best of our knowledge, this strong correlation between electrical and morphological characterization is unreported in previous studies of MCBJs and EBJs.

**Demonstration of molecular junctions formed using CDBJ**. To test whether CDBJs are suitable for performing electrical transport experiments on molecules, we deposited oligo(phenylene ethynylene) with acetyl-protected thiol groups (OPE3-SAc) from solution immediately after electrical pre-characterization to identify tunneling junctions (see Methods for more details on the device fabrication). OPE3 is a widely studied, conjugated 'reference' molecule with a length of about 1.8 nm[29–31]. Figure 5 shows an idealized OPE3 molecular junction and the $I$–$V$ characteristic of a tunneling gap before and after molecule deposition recorded under ambient conditions. An increase in conductance from $2 \times 10^{-6}$ $G_0$ to $1 \times 10^{-4}$ $G_0$ can be observed. A similar increase in conductance was found in 6 out of 13 tunneling junctions investigated in this study (see Supplementary Fig. 5), where the logarithmic conductance values group around $2.4 \times 10^{-4}$ $G_0$ after deposition. This value is close to the values of 1 to $3 \times 10^{-4}$ $G_0$ found in MCBJ measurements on OPE3-SAc[32,33]. The variation in conductance values can be explained by different couplings between the gold contact and the OPE3 molecule or the formation of parallel molecular junctions inside the gap. The $I$–$V$ histogram in Fig. 5 which consists of 64 individual $I$–$V$ traces shows low variability reflecting the high stability of the junction. However, in some devices telegraph noise is observed which can be attributed to molecular rearrangements and the formation of multiple junction configurations[34] (see Supplementary Fig. 5d–f).

It is worth mentioning that CDBJ based molecular junctions can be operated at cryogenic temperatures. By cooling down the junctions from 300 K to 7 K, we found a small decrease in conductance (see Fig. 6) and no device failure in all six tested junctions (Supplementary Fig. 5). This high device stability could allow for future detailed inelastic tunneling spectroscopy studies

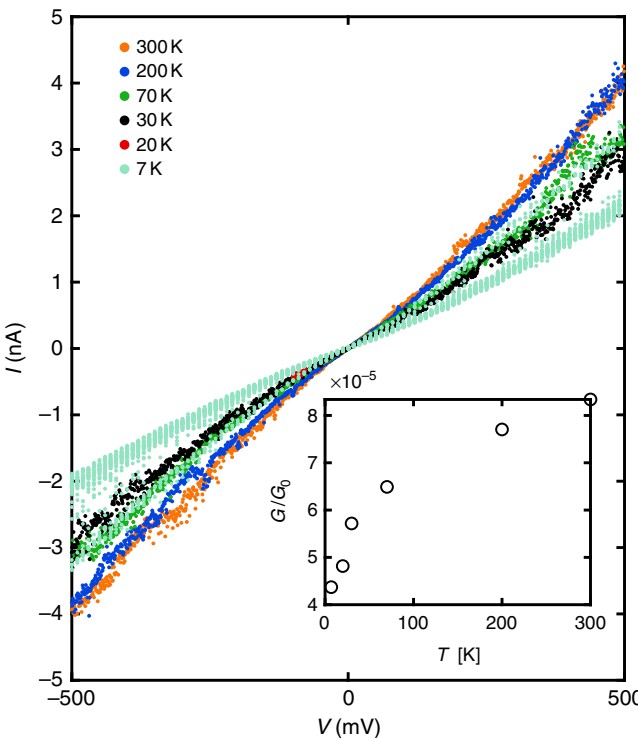

**Fig. 6** Temperature dependent *I–V* characteristics of a molecular junction. The *I–V* characteristics of an OPE3 junction in vacuum reveals the suitability and stability of molecular junctions formed using CDBJ for experiments from 300 K down to cryogenic temperatures. The low bias conductance (*G*) of the molecular junction decreases with decreasing temperature (inset)

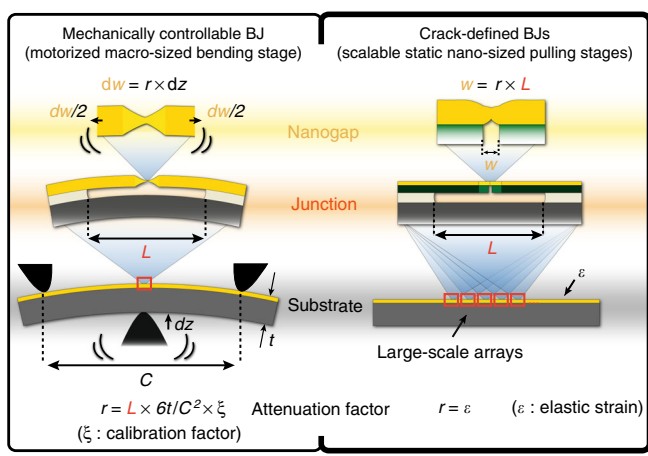

**Fig. 7** Cross-sectional schematics of a MCBJ integrated in a 3-point bending stage and of CDBJs. The dynamically controllable mechanically controllable break junction (MCBJ) allows for continuous monitoring of the breaking process and tuning of the resulting inter-electrode separation. However, the need to maintain a precise substrate curvature restricts the usability to single junctions. In contrast, a crack-defined break junction (CDBJ) has a fixed pulling action, but the self-breaking process triggered by crack formation and retraction of the cantilevers is highly parallelizable and can be applied to large-scale arrays of many thousands of break junctions simultaneously. Further, unlike in a MCBJ 3-point bending stage, the attenuation factor of a CDBJ is known prior to breaking as it is equal to the elastic strain of the bridge material. It is worth noting that any CDBJ fabricated in this work can also be integrated in a 3-point bending stage that, assuming a typical bridge geometry, has an attenuation factor $r \sim 6Lt/C^2 = 6 \times 1 \times 10^{-6} \times 5 \times 10^{-4}/(2.5 \times 10^{-2})^2 \approx 5 \times 10^{-6}$

of vibrational[34] or many-body effects[35] in single-molecule junctions.

## Discussion

In terms of the breaking process, a CDBJ is analogous to a MCBJ since both accomplish the breaking of the metal constriction through the application of a controlled pulling force, as illustrated in Fig. 7. In terms of applicability, a CDBJ is closer to an EBJ as they both are suitable for producing molecular junctions where the gaps between the contact electrodes do not have to be reconfigured. However, electromigration of pre-fabricated metal constrictions forms EBJs at a rate of one junction in a few minutes at best, and if not carefully controlled with active feedback, can generate undesirable debris in the vicinity of the created gap[36,37]. In contrast, more than 20 million of debris-free break junctions can be formed on a single substrate with the CDBJ methodology in about 5 h using wafer-level processing (including thin film deposition, patterning and release-etching on a 100 mm diameter wafer), considering a very conservative junction footprint of 400 μm². This is equivalent to producing approximately one junction every 1 ms, thereby improving fabrication throughput more than $10^5$-fold, while typical fabrication yields are comparable for both methodologies[5,16,38]. As a first step towards establishing the utility of our approach, we have demonstrated the viability to form molecular junctions using CDBJs, and also their compatibility with cooling down to liquid helium temperatures.

The yield of tunneling devices in the present study is likely limited by the combined effect of the nanocrystallinity of the Au and TiN films, and the atomic scale bluntness of the notches used to localize the crack-formation. The variability in the

orientation of cracks formed in TiN and the variability of cracked-edge recession in TiN, translates into variable straining of the Au film on top, whose nanocrystallinity further compounds the observed variations in the final outcome for nominally identical bridges. In its ideal manifestation, our approach would combine a single-crystalline insulating brittle cracking layer with a single-crystalline conductive ductile electrode layer on top, to eliminate material variability. Restricting the discussion to the material system used in the present study, the most promising route to improving yield is by increasing the grain size of the evaporated Au film through various handles such as decreasing the Au deposition rate, heating the substrate during Au deposition, or by annealing the deposited films after deposition[39–41]. Our CDBJ methodology paves the way towards the long-term goal of molecular electronics, namely the integration of molecular functionalities into electrical circuits and systems consisting of dense arrays of interconnected molecular junctions. In view of the sheer number of break junctions produced in a single batch, the CDBJ methodology drastically reduces processing time for the fabrication of individual break junctions and provides a platform for investigations of electrical, mechanical, thermal, and optical properties of molecules and atomic-sized contacts on a statistically significant number of junctions. Furthermore, since the general CDBJ methodology is compatible with CMOS wafers, CDBJs can be integrated on top of CMOS circuits. With such an approach, each junction could be individually connected to, and addressed by conventional solid-state integrated electronic circuits (ICs). Finally, the significance of this work goes beyond nanogap electrodes made of gold since the CDBJ methodology can be extended to other classes of materials by substituting gold with any electrode material that exhibits interesting electrical, chemical, and

plasmonic properties for applications in molecular electronics and spintronics, nanoplasmonics, and biosensing.

## Methods

**Wafer preparation**. A 100 mm diameter, 525 μm thick p-doped single-crystalline silicon wafer (100) was used as a starting substrate. A 100 nm thick silicon oxide layer (SiO$_2$) was thermally grown on the silicon wafer by wet oxidation. Then, a 200 nm thick layer of amorphous silicon (a-Si) was deposited using PECVD (Applied Materials Precision 5000 Etcher, at a chamber pressure of 3 Torr, a temperature of 400 °C, and RF power of 25 W using a mixture of silane (SiH$_4$) at 300 sccm flow, and nitrogen (N$_2$) at 300 sccm flow). Then, a 50 nm thick layer of titanium nitride (TiN) was deposited by atomic layer deposition (Beneq TFS 200) at a temperature of 350 °C in 2700 cycles of titanium tetrachloride (TiCl$_4$) (pulse time 150 ms, purge time 500 ms) and ammonia (NH$_3$) (pulse time 1 s, purge time 1 s) as precursors. Next, a 3 nm thick chromium (Cr) adhesion layer and a 10 nm thick gold (Au) layer were evaporated on top of the TiN.

**Fabrication of crack-defined break junctions**. The notched bridges with connected probing pads were first defined in a resist mask (SPR-700) using a projection stepper system (Nikon NSR TFHi12 I-line Stepper, dose 190 mJ/cm$^2$). While the stepper has a nominal resolution limit of about 0.5 μm, we could make notched constrictions as narrow as 90 nm using a triangular bridge design (see details in Supplementary Fig. 3) in combination with a slight over-exposure of the photoresist. The photoresist pattern was transferred into the Au/Cr/TiN layer stack by a combination of argon ion beam etching (Oxford Instruments, Ionfab 300 plus) to sputter-etch the gold and Cr layers, and an anisotropic plasma etch (Applied Materials, Precision 5000 Etcher, at a chamber pressure of 200 mTorr and an RF power of 600 W using a mixture of BCl$_3$ at 40 sccm flow, Cl$_2$ at 15 sccm flow, CF$_4$-O$_2$ at 15 sccm flow, N$_2$ at 15 sccm flow for 25 s) to pattern the TiN layer. The resist mask was subsequently removed with a remover (Microposit remover 1165) at 50 °C in an ultrasonic bath for 5 min. All bridges were released in a single step by sacrificial isotropic etching of the a-Si using an inductively coupled plasma (STS ICP DRIE, at a chamber pressure of 10 mTorr and an RF coil generator power of 300 W in SF$_6$ at 160 sccm flow) for 56 s leading to an undercut length of 600–800 nm, depending on the location on the wafer. A sacrificial release using dry etching rather than wet etching was favored as it was observed that the gold layer quickly lost its adhesion to TiN upon contact to common wet etchants used to etch aluminum oxide, which was the sacrificial layer used in earlier versions of the crack-junction process[25,26], thereby preventing the pulling action of the TiN cantilevers to be readily imparted to the gold.

Although the gold layer was thin, it was possible to obtain stable electrical contacts between the probe needles and the probing pads connected to the junctions. The 50 nm thick layer of electrically conducting TiN below the gold layer served as mechanical and electrical supporting layer. To prevent electrical currents leaking though the TiN and the Cr layers at the cracked extremities of the cantilevers, short selective etches were applied successively to retreat the TiN layer and the Cr layer at the junctions. The etching was done by immersing the wafer in a SC-1 solution (1(H$_2$O$_2$):1(NH$_4$OH):5(H$_2$O)) for 30 min at room temperature, blow drying in N$_2$ and subsequently placing the wafer in oxygen plasma in the ICP (at a chamber pressure of 40 mTorr and an RF coil generator power of 300 W in O$_2$ at 49 sccm flow) for 2 min. In these etch-steps, TiN and Cr were etched isotropically by 20 nm and 3 nm, respectively. Along with Cr etching, the oxygen plasma played the role of pre-cleaning the CDBJs before the electrical characterization. Direct visual confirmation of the TiN undercut was obtained by SEM imaging of fully-released and overturned gold-coated TiN structures (see Supplementary Fig. 6).

Note that etching in SC-1 solution did not affect the adhesion of Cr/Au on TiN. Moreover, since the cantilevers with designs that are optimal for CDBJ formation are sufficiently stiff to resist stiction, no critical point drying was used in our process. Although titanium (Ti) is as effective an adhesion layer as Cr for CDBJ fabrication, Cr is preferred because it can be etched in an oxygen plasma to eliminate possible parasitic conduction paths through the adhesion layer after CDBJ formation.

**Electrical characterization**. Electrical two-point measurements were favored over four-point measurements as it was found in initial characterization experiments of CDBJs that four-point probing bore an increased risk of inducing electrical damage to the junctions. For the first set of 90 probed junctions (see Supplementary Fig. 3), SEM imaging was performed only after the electrical characterization to eliminate the possibility of damage and carbon contamination induced by electron beam irradiation. Due to the lack of morphological information prior to the electrical experiments, the conductance mechanism of the junctions was unknown prior to probing. Thus, each junction was characterized following a rigorous procedure to minimize risks of exposing junctions to unnecessarily high voltage or current levels during electrical characterization. In a first step, the CDBJ was characterized for low resistance ohmic behavior. This was done by sweeping the voltage from 0 to 20 mV in steps of 0.5 mV at a current compliance of 50 μA. If the current remained within the noise level, the junction was characterized at higher voltages. This was done by sweeping the

voltage from 0 to 10 V in steps of 0.5 V and at a current compliance of 3 pA. If currents above the noise level of hundreds of fA could be detected, a final sweep was performed. This was done by raising the current compliance to 500 pA and adjusting the voltage sweep according to the voltage detected at the moment of reaching the previously set current compliance of 3 pA. The same electrical probing procedure was also applied to the second set of junctions (see Supplementary Fig. 4), a subset of which were first identified using SEM as likely to be tunneling and then electrically characterized.

**Morphological characterization**. The morphology of the CDBJs was characterized by SEM imaging (Zeiss Gemini Ultra 55). To obtain sharp and high-resolution images of the nanogaps, we used an aperture size of 30 μm, a magnification of ×150k, an acceleration voltage of 3.5 keV, a working distance of 4 mm, a resolution of 1024 × 768 and a noise reduction based on line average with a line average count of 42. A series of successive SEM images of four junctions of different types presented in Supplementary Fig. 7 show that these imaging conditions do not cause any noticeable change in the morphology of the junctions. The 1250 CDBJs examined to estimate the yield of fracture were inspected using an aperture size of 30 μm, magnification of ×100k, acceleration voltage of 4 keV, working distance of 5.5 mm, resolution of 1024 × 768 and a noise reduction based on line average with a line average count of 42.

**1-D transmission model for tunneling I–V characteristics**. The I–V characteristics of tunneling devices in our study were fitted to a 1-D model for a single transmission channel across a tilted trapezoidal barrier. This 1-D version of Simmons model was reported by Mangin et al.[28], and can describe the asymmetry, as well as the regimes of direct and field-emission tunneling in I–V characteristics. Our implementation is identical to Mangin et al., and we reproduce the relevant equations below for the sake of clarity.

$$I(V) = \frac{2e}{h} \int_0^\infty [f(E) - f(E - eV)] T(E, V) dE, \qquad (2)$$

where $f$ is the Fermi distribution, and $T(E,V)$ is the transmission probability of an electron through the potential barrier. In the Wentzel Kramers Brillouin (WKB) approximation, the transmission probability is given by

$$T(E, V) = \exp\left\{ -\int_{z1}^{z2} \frac{4\pi}{h} \sqrt{2m[\varphi(z, V) - E]} dz \right\}. \qquad (3)$$

Here, $\varphi(z,V)$ is the potential profile along the tunneling gap (which is along the z direction), and z1 and z2 are the solutions of the equation $\varphi(z,V)-E = 0$. The potential profile of the trapezoidal barrier along the direction z is represented in terms of the work functions of the left ($\varphi_L$) and right ($\varphi_R$) electrodes, and the gap width ($d$) as

$$\varphi(z, V) = \varphi_L + (\varphi_R - \varphi_L - eV)\frac{z}{d}. \qquad (4)$$

The fitting of Eq. 2 to I–V data was performed in MATLAB using the non-linear least squares solver. When data was only collected for positive bias voltages, as was the case for all the junctions not used in the molecular junction experiments in our study, we found that assuming a symmetric barrier ($\varphi_L = \varphi_R$) produced better fits. While both symmetric and asymmetric barrier could be fitted to the I–V data to produce fits of visually similar quality (see Junction 3, 7, 8, 13, and 22 in the data series presented in Supplementary Note 1), the values for barrier height for the case of symmetric barrier were more reasonable; when an asymmetric barrier was assumed, the two barrier heights were found to be comparable in most cases, but were very different in a few cases (e.g., 0.6 and 3.9 eV or 0.05 and 2.8 eV). The gap widths did not vary by more than 0.1 nm between the two barrier models unless the barrier heights were very different for the left and right electrodes like in the cases mentioned above. Even when this is so, the maximum difference in gap widths is always less than 0.6 nm. We therefore only present the fit parameters for the symmetric barrier model in Supplementary Table 1 and use the gap widths determined using the symmetric barrier in all our discussions in the manuscript.

**Fabrication and characterization of molecular junctions**. To test the formation of molecular junctions, we only chose CDBJs which displayed measurable tunneling currents for applied bias below 500 mV. By systematically sweeping 540 devices from 6 cells on the wafer we identified 13 such tunneling junctions. Ten of these tunneling junctions were from a subset of 180 devices fabricated using the most promising design parameters described in Results ($w = 9$ nm, determined by a bridge length of $L = 3.3$ μm, and a 90 nm wide constriction). The remaining 3 junctions were from adjacent columns in the cells with sub-optimal designs, thus explaining the lower yield of tunneling junctions obtained for this other subset of devices.

The CDBJs were cleaned by soaking them in dichloromethane (DCM) prior to molecule deposition. A 1 mM solution of OPE3-SAc in DCM was mixed with two equivalents of tetrabutylammonium hydroxide dissolved in DCM[32] and then immediately drop cast on the CDBJs. To remove molecules not bound to the gold electrodes the samples were soaked in DCM for 5 min after molecule deposition followed by blow drying with $N_2$.

$I$–$V$ curves before and after molecule deposition on the 13 previously identified junctions were recorded at room temperature in open atmosphere in a Lake Shore cryogenic probe station using home-built low-noise DC electronics. The $I$–$V$ histograms after molecule deposition consist of 64 consecutive $I$–$V$ measurements. Low temperature characterizations were performed in vacuum in the same probe station equipped with a liquid helium flow cryostat.

**Single-level model for molecular conduction**. In the zero-temperature limit the transport through a conjugated molecule can be modeled by a Breit–Wigner single-level model:

$$I(V) = \frac{G_0}{e} \Gamma \left[ \arctan\left( \frac{\varepsilon_0 + \frac{1}{2}eV}{\Gamma} \right) - \arctan\left( \frac{\varepsilon_0 - \frac{1}{2}eV}{\Gamma} \right) \right] \qquad (5)$$

where $\Gamma = \Gamma_L + \Gamma_R$ is the total tunnel coupling, $\varepsilon_0$ is energy of the single level, and $G_0 = \frac{2e^2}{h}$ is the quantum of conductance. By fitting the low-temperature $I$–$V$ curves recorded at $T = 7$ K for $-0.5$ V$<V_{bias}<0.5$ V of OPE3 molecular junctions using Eq. 5, we can extract $\Gamma$ and $\varepsilon_0$ for all six devices (see Supplementary Table 2). These extracted parameters are very similar to values found in MCBJ measurements on OPE3-SAc[32]. In addition, the strong tunnel couplings $\Gamma \gg k_B T$ verifies the use of the single-level model.

**Data availability**. The data that support the findings of this study are available from the corresponding authors upon reasonable request.

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

## Acknowledgements

This work was supported by the Swedish Research Council (Grant No. 2016-04852) and the European Research Council (Grant No. 277879 and No. 812975). P.G. acknowledges a Marie Skłodowska-Curie Individual Fellowship under grant Ther-SpinMol (ID: 748642) from the European Union's Horizon 2020 research and innovation programme. The authors thank D. Stefani for help with the preparation of the molecule solutions.

## Author contributions

V.D. designed the concept, carried out the fabrication of the wafer, as well as the morphological and electrical characterization of bare junctions, and wrote most of the paper. S.N.R. contributed to writing the paper and to the analysis and interpretation of the experimental data. P.G. and S.C. supervised by H.S.J.Z. carried out the fabrication and electrical characterization of molecular junctions, analyzed the data and wrote that part of the manuscript. F.N. and G.S. provided guidance and supervised the work. All the

authors discussed the results. V.D., S.N.R., and P.G. prepared the final manuscript with comments from all authors.

## Additional information

**Competing interests:** The authors declare no competing interests.

