## [Peer Review File · Nature Communications]

Reviewer #1 (Remarks to the Author):

The authors report a wafer-scale approach to fabrication of nanoscale gaps between notched (patterned constriction) extended Au electrodes. The fabrication protocol employs a sacrificial amorphous silicon layer under a highly tensilely stressed TiN layer, with the top layer being Cr(adhesion)/Au electrode material. Dry etching removes the a-Si, undercutting the strained TiN, which cracks at the constriction and retracts, separating the electrodes. With proper choice of geometric parameters, the process may be tuned to produce nanogaps rather than electrodes still connected by stretched metal filaments. Further etching is required to ensure that there are not parasitic conducting paths due to the TiN underlayer.

The results of this process are very impressive, and I find the paper to be clearly presented. This is a clear advance in technique. I think the paper is appropriate for publication. I do think that the paper could be improved a bit if the authors addressed some minor questions:

1) I think the paper would benefit from a slightly more extensive discussion of process control and variability. Do the authors think that with sufficient process control it would be possible to improve the yield of devices with detectable tunneling? Is it the microscopic granularity and morphology of the Au film that the authors believe is the limiting factor here in terms of device variability?

2) Have the authors tried Ti as an adhesion layer rather than Cr? It seems to me that this could promote better adhesion with the underlying TiN.

3) Regarding the issue of adhesion, I was confused by one aspect of the process. The authors explicitly state that they use a dry etching approach to free the TiN/Cr/Au constriction, because wet etching seemed to hurt the Cr/Au adhesion on the TiN. However, in the next paragraph (near the top of page 6), the authors indicate that a key subsequent step is the removal of residual TiN and Cr by wet etching (SC-1 solution), followed by oxygen plasma. Should I infer, then, that the SC-1 etch does not hurt the Cr/Au adhesion? Also, were there any issues with surface tension or capillary forces damaging the suspended structures? I take it that supercritical drying was not needed.

4) It is probably worth mentioning in the introduction that there have been some approaches with self-aligned techniques to produce very small interelectrode gaps in potentially wafer-scalable means. See Tang, et al., *Microelectronic engineering* 83, 1706-1709 (2006); Fursina et al., *Applied Physics Letters* 92, 113102 (2008); Zhu et al., *Small* 7, 1761-1766 (2011).

Reviewer #2 (Remarks to the Author):

This is a somewhat exceptional paper, because it does not report any new science. It reports on a new wafer-scale fabrication method for obtaining many sets of electrode pairs separated by nano-scale gaps. In research on single-molecule transport such bridges are commonly formed by mechanical break junctions (MCBJ), by scanning tunneling microscopes (STM), or by electromigration breaking (EBJ). The advantage of this new method, that employs the built-in strain in a TiN layer for breaking of a gold bridge, is that it permits producing massive numbers of junctions in parallel.

Yet, what have we learned? The process is not obviously ground breaking. For research, the mechanical and STM-based methods produce much higher numbers of useful junctions, because a new junction can be formed by indentation and retraction on sub-second time scales. Electromigration junctions are usually fabricated with orders of magnitude smaller numbers of structures on a chip, typically 100. Still, this is sufficient because the success rate is comparable to the crack-defined break junctions (CDBJ) reported here, about 8%, as acknowledged by the authors. This gives several successful EBJ junctions on a chip. The strong point of the EBJ, that is presumably shared by the CDBJ, is that they are solid-state and can be cycled to cryogenic temperatures and high magnetic fields without losing contact. A successful junction cooled to helium temperature is typically studied for days. This is the bottle neck, and it does not help to have thousands times more junctions for testing. One could, of course, limit the investigations to simple characteristics at room temperatures, but then the competition of the MCBJ and STM methods will win over the CDBJ.

In terms of applications, it is true that the fabrication method is compatible with CMOS and permits integrating the junctions into electronics devices. However, here the yield of 8% of useful junctions is not very attractive, and there is no obvious route for improving this. Building networks of interconnected molecules will not offer a great advantage, given the relatively large footprint per junction.

Nevertheless, the method is original and I have some sympathy for this work. Who knows what it may bring? In my view the decision on publication depends on the policy of the editor since the work does not contain any new scientific understanding. A demonstration of experiments on a molecule bridging the junctions would have been more convincing.

The method is clearly described and the paper is well-organized. There is one point for improvement: the estimates of the gap distance are based on application of the Simmons formula for tunneling, using the Au work function of 5.1eV. However, this is the value for clean Au in

vacuum. Since these experiments are performed in ambient, a much smaller value must be used. One typically encounters values of 1 eV, depending on coverage of the surface by adsorbents.

Reviewer #3 (Remarks to the Author):

Report about manuscript: "Massively parallel fabrication of gold breakjunctions featuring sub-3 nm gaps" by Dubois et al.

The authors report a fabrication method for a large number of metallic nanogaps with yield of around 8% and relatively high integration density. The work is interesting and might be published, but I have several remarks.

1. I find the term breakjunction misleading. It is true that the nanogaps have been fabricated by breaking a formerly continuous wire, but this is not important.
2. The term break junction implies that the junction is formed while breaking and is tunable and is not applied to prebroken nanogaps. As the authors state correctly electromigrated nanogaps are called breakdown junctions and not break junctions.
3. There is one important variant of break junctions which is not mentioned at all, namely STM break junctions. They are at least as common for contacting molecules as MCBJs and EBDJs. They are correctly named break junctions because of the in situ formation of the molecular junction.
4. Besides the tunability the in situ formation of the molecular junctions is important because freshly broken and therefore clean metal surfaces are formed upon each breaking. Fresh metal surfaces are important for the formation of molecular junctions.
5. The authors claim that their method is suitable for forming molecular junctions but they do not give any evidence for this claim. They show that with a certain yield they can form tunnel contacts. The spread of the tunnel width is indicated with 1.2 to 3 nm. This spread is much too much to form single-molecule junctions with reasonable distribution of conductance. The authors must at least give one demonstration of a successful formation of a molecular junction.
6. The authors claim to see Fowler-Nordheim tunneling. From the data given in the SI it is not clear whether the tunneling is Fowler-Nordheim type. In order to support the claim they should check their data in a Fowler-Nordheim representation. To come into the Fowler-Nordheim regime the applied voltage must exceed the tunnel barrier height. The authors indicate to have used the bulk value for surfaces in vacuum (~ 5 eV). The applied voltage in the cases shown is always below 5 V. Thus, they cannot prove whether it is Fowler-Nordheim or not.

7. The authors say to have used the Simmons model there are so many Simmons models around, the authors should indicate which approximation, which assumption for the tunnel barrier they used.
8. The gap width has been determined from the Simmons model using the value ~ 5 eV for the work function. This is highly questionable for ligament –formed structures that have been exposed to atmosphere, as is shown e.g. PHYSICAL REVIEW B 80, 235432 2009 written by the authors of their reference 25.
9. The Simmons model (in the usual variation for a rectangular barrier) contain three fitting parameters, which are not independent of each other: the work function the tunnel gap and the cross section. Giving the limited voltage window I assume that there are several combinations of these three parameters which give rise to fits of comparable quality. A basic quality check is to use also the opposite polarity of the voltage bias and verify the symmetry. In the reference given above a decent analysis of tunnel data with a one-dimensional model is given. This model contains for symmetric electrodes (symmetric IVs) only two parameters. It is worth testing this model.
10. I wonder whether the nanogap structures keep their geometry during SEM inspection. To achieve the high spatial resolution the electron density of the imaging electron beam is usually relatively high. From own experience I know that motion of atoms takes place even on the several nanometer scale. It might be that because of the relatively low acceleration voltage used here the bombardment is less invasive than with usual SEMs, but the authors should give evidence for the noninvasiveness of their characterization method. One possibility would be to vary the imaging current and check the stability of the images.
11. –I wonder what the progress is compared to their reference 27. It is mentioned in the reference list but I do not see where it is quoted in the manuscript.

Please note that in all our responses below, the page and line numbers mentioned refer to those in the highlighted versions of our revised manuscript and Supplementary Information.

Reviewers' comments:

Reviewer #1 (Remarks to the Author):

Reviewer comment:

The authors report a wafer-scale approach to fabrication of nanoscale gaps between notched (patterned constriction) extended Au electrodes. The fabrication protocol employs a sacrificial amorphous silicon layer under a highly tensilely stressed TiN layer, with the top layer being Cr(adhesion)/Au electrode material. Dry etching removes the a-Si, undercutting the strained TiN, which cracks at the constriction and retracts, separating the electrodes. With proper choice of geometric parameters, the process may be tuned to produce nanogaps rather than electrodes still connected by stretched metal filaments. Further etching is required to ensure that there are not parasitic conducting paths due to the TiN underlayer.

The results of this process are very impressive, and I find the paper to be clearly presented. This is a clear advance in technique. I think the paper is appropriate for publication. I do think that the paper could be improved a bit if the authors addressed some minor questions:

Our response:

We would like to thank the reviewer for the clear and succinct analysis of our work and for the favorable assessment.

Reviewer comment:

1) I think the paper would benefit from a slightly more extensive discussion of process control and variability. Do the authors think that with sufficient process control it would be possible to improve the yield of devices with detectable tunneling? Is it the microscopic granularity and morphology of the Au film that the authors believe is the limiting factor here in terms of device variability?

Our response:

The formation of crack-defined gold break junctions is a two-step process. First, the removal of the sacrificial layer causes a redistribution of internal stress in the TiN layer, and the stress concentration at the pre-designed notch localizes the crack formed by the brittle fracture of the TiN. Next, as the cracked edges of the TiN cantilevers recede, the Au layer on top is stretched across the gap, and eventually undergoes ductile fracture if pulled beyond the breaking point. Since both Au and TiN are nanocrystalline films, and the notches are far from being atomically sharp, a significant variation in the orientation of crack formation in the TiN with respect to the notch and the magnitude of cracked-edge recession in the TiN cantilever results. This translates into asymmetric straining of the Au film on top; the non-uniformity caused by the nanocrystallinity in the Au film in turn produces significant variations in the final outcome. We do believe, as the reviewer suggests, that any effort to improve tunneling yield further than we have achieved will start by controlling the grain sizes in the Au films, for instance, by changing the rate of deposition, the substrate temperature during deposition or by annealing after deposition.

In the revised manuscript we have included a paragraph in the Discussion section (page 6, line 201–210) to address this topic.

Reviewer comment:

2) Have the authors tried Ti as an adhesion layer rather than Cr? It seems to me that this could promote better adhesion with the underlying TiN.

Our response:

This is an interesting point. Proper choice of the adhesion layer is indeed an important aspect of the CDBJ methodology without which mechanical pulling of the gold on TiN would not work.

As the reviewer suggests, since Ti is an element that is also present in TiN, Ti could be better than Cr as an adhesion layer of gold on TiN. In fact, we have already tried using Ti and found it worked well with the CDBJ technique. Moreover, we also found that Ti is compatible with the SC-1 etch used to etch TiN and avoid parasitic conducting paths. However, since we did not find any significant difference between Ti and Cr from an adhesion perspective, we chose Cr instead of Ti, since Cr can be etched in an oxygen plasma. This property was used to clean the junctions before electrical characterization, to ensure that there were no parasitic conduction paths through the adhesion layer, as already mentioned in the manuscript (page 7, line 256–262). We have now added a sentence in the revised manuscript to explicitly state the above-mentioned reason for our preference for Cr (page 7, line 268–270).

Reviewer comment:

3) Regarding the issue of adhesion, I was confused by one aspect of the process. The authors explicitly state that they use a dry etching approach to free the TiN/Cr/Au constriction, because wet etching seemed to hurt the Cr/Au adhesion on the TiN. However, in the next paragraph (near the top of page 6), the authors indicate that a key subsequent step is the removal of residual TiN and Cr by wet etching (SC-1 solution), followed by oxygen plasma.

Our response:

As correctly identified by the reviewer, our statements regarding the issue of adhesion was indeed confusing, since we used an SC-1 etch, which is a wet etch, to selectively remove TiN after forming crack junctions by dry etching.

Our comment in the original manuscript regarding the preference for dry etching to form the crack-defined gold break junctions comes from our previous abortive attempts to use aluminium oxide as the sacrificial layer in combination with wet etching (in KOH or aluminium etch) to trigger crack formation. However, this resulted in loss of adhesion of the Au film on TiN. In order to avoid this from occurring, we switched to using amorphous silicon as the sacrificial layer and dry etching to release the notched TiN/Cr/Au structure. This is the material combination we have reported in the manuscript, without the back story.

We have now clarified this issue in the revised manuscript (page 7, line 250-251).

Reviewer comment:

Should I infer, then, that the SC-1 etch does not hurt the Cr/Au adhesion?

Our response:

Yes, it is true that SC-1 etch does not cause the loss of Cr/Au adhesion on TiN. We have added a statement in the revised manuscript clarifying this point (page 7, line 266).

Reviewer comment:

Also, were there any issues with surface tension or capillary forces damaging the suspended structures? I take it that supercritical drying was not needed.

Our response:

The use of the SC-1 etch, which is a wet etch, indeed poses the problem of stiction of suspended structures while drying. We observed stiction of long CDBJs, however, the CDBJs with designs that are optimal for break junction formation were sufficiently stiff to resist stiction. We did not use critical point drying in our study. In our additional work performed to demonstrate the formation of molecular junctions with CDBJ, we deposited molecules by immersing our devices in a solution containing the molecules, then rinsing the devices with a solvent and finally blow dried the devices with N₂, and almost all previously tunneling junctions were still active after this process. We show the I-V characteristics of six such junctions before and after molecule deposition in Supplementary Figure 8 (SI page 23–24). We have now added a sentence to the revised manuscript to state that we did not use critical point drying in our process because the cantilevers were sufficiently stiff to resist stiction, and also explicitly state that we blow dried the devices in N₂ after the SC-1 etch (page 7, line 259; page 7, line 266–268).

Reviewer comment:

*4) It is probably worth mentioning in the introduction that there have been some approaches with self-aligned techniques to produce very small interelectrode gaps in potentially wafer-scalable means. See Tang, et al., *Microelectronic engineering* 83, 1706-1709 (2006); Fursina et al., *Applied Physics Letters* 92, 113102 (2008); Zhu et al., *Small* 7, 1761-1766 (2011).*

Our response:

We thank the reviewer for bringing these works to our attention. We have cited these works in the “Introduction” section of our manuscript. However, we must contend that while these techniques are potentially scalable methods of producing nanoscale slits, we cannot see a direct route for parallel fabrication of nanogaps with junction cross sectional area $\sim 1\text{-}10\text{ nm}^2$ typical of tunneling junctions formed by MCBJ, EBJ or CDBJ, without using ebeam lithography (which is a serial process) to define at least one narrow electrode.

In the revised manuscript we have cited the suggested papers to present a more complete picture of previously reported potentially scalable approaches to fabricate nanogaps (page 1–2, line 15–20).

Reviewer #2 (Remarks to the Author):

Reviewer comment:

This is a somewhat exceptional paper, because it does not report any new science. It reports on a new wafer-scale fabrication method for obtaining many sets of electrode pairs separated by nano-scale gaps. In research on single-molecule transport such bridges are commonly formed by mechanical break junctions (MCBJ), by scanning tunneling microscopes (STM), or by electromigration breaking (EBJ). The advantage of this new method, that employs the built-in strain in a TiN layer for breaking of a gold bridge, is that it permits producing massive numbers of junctions in parallel.

Yet, what have we learned? The process is not obviously ground breaking. For research, the mechanical and STM-based methods produce much higher numbers of useful junctions, because a new junction can be formed by indentation and retraction on sub-second time scales. Electromigration junctions are usually fabricated with orders of magnitude smaller numbers of structures on a chip, typically 100. Still, this is sufficient because the success rate is comparable to the crack-defined break junctions (CDBJ) reported here, about 8%, as acknowledged by the authors. This gives several successful EBJ junctions on a chip. The strong point of the EBJ, that is presumably shared by the CDBJ, is that they are solid-state and can be cycled to cryogenic temperatures and high magnetic fields without losing contact. A successful junction cooled to helium temperature is typically studied for days. This is the bottle neck, and it does not help to have thousands times more junctions for testing. One could, of course, limit the investigations to simple characteristics at room temperatures, but then the competition of the MCBJ and STM methods will win over the CDBJ. In terms of applications, it is true that the fabrication method is compatible with CMOS and permits integrating the junctions into electronics devices. However, here the yield of 8% of useful junctions is not very attractive, and there is no obvious route for improving this. Building networks of interconnected molecules will not offer a great advantage, given the relatively large footprint per junction.

Nevertheless, the method is original and I have some sympathy for this work. Who knows what it may bring? In my view the decision on publication depends on the policy of the editor since the work does not contain any new scientific understanding. A demonstration of experiments on a molecule bridging the junctions would have been more convincing.

Our response:

We thank the reviewer for sharing the frank perspective of our work, and for appreciating its originality. As the reviewer suggests, while the relatively low yield currently stymies the direct integration of CDBJ with CMOS-based electronic circuits, we hope that it motivates further research that could increase yield enough to make this a viable option. Based on our observations, we think that the low yield is a consequence of the variability in the final outcome of nominally identical bridges caused by the nanocrystallinity of the TiN and Au films. Therefore, the ideal system to eliminate such variability introduced by the materials and make a significant jump in yield would combine a single crystalline/large grain brittle insulator (instead of TiN) and a single crystalline/large grain ductile conductor (instead of evaporated Au). We have discussed this aspect in the “Discussion” section of the revised manuscript, in combination with a response to a related comment (#1) by Reviewer 1 (page 6, line 201–210).

Following the suggestion of Reviewer’s 2 and 3 (in the next remark), we have carried out new experiments to demonstrate the formation of molecular junctions using our CDBJ. The molecule we use is oligo(phenylene ethynylene) with acetyl protected thiol groups (OPE3-SAc). OPE3 is a widely studied conjugated reference molecule. Of 13 new devices identified to be tunneling before

OPE3 deposition, we formed molecular junctions in six of them. The formation of molecular junctions is indicated by a large increase in low bias conductance of the junctions after molecular deposition: in five of six cases this increase in conductance was more than one order of magnitude. Moreover, the logarithmic conductance values group around $2.4 \times 10^{-4}G_0$ after deposition, which is close to the values of $1-3 \times 10^{-4}G_0$ found in MCBJ measurements on OPE3-SAc.

As correctly predicted by the reviewer, our CDBJ could be cooled down to cryogenic temperatures (7K), with none of the six molecular junctions failing during the process.

We have added a new section titled “Demonstration of molecular junctions formed using CDBJ” to the revised manuscript where these experiments are described (page 5, line 165–184). A section in the methods section of the revised manuscript titled “Fabrication and characterization of molecular junctions” describes the process of molecule deposition and the related electrical characterization (page 8, line 299–307). Figure 4 in the revised manuscript presents a schematic of the molecular junction and an I-V histogram of one molecular junction. Supplementary Figure 8 presents the I-V histograms under ambient conditions for all six molecular junctions, alongside the I-V curves before molecule deposition, and the I-V curves of all six junctions at 7K (SI page 23–24). In Supplementary Figure 9 we present the I-V characteristics of one molecular junction at six different temperatures: 300, 200, 70, 30, 20 and 7K (SI page 25). We have also added the phrase “for molecular devices” to the title to reflect the fact that CDBJ could be used to form molecular junctions.

Reviewer comment:

The method is clearly described and the paper is well-organized. There is one point for improvement: the estimates of the gap distance are based on application of the Simmons formula for tunneling, using the Au work function of 5.1eV. However, this is the value for clean Au in vacuum. Since these experiments are performed in ambient, a much smaller value must be used. One typically encounters values of 1 eV, depending on coverage of the surface by adsorbents.

Our response:

In our original submission we used the 3D form of Simmon’s model for a symmetric rectangular barrier, using the approximate analytical expressions reported in Simmon’s original paper for tunneling current densities across a symmetric potential barrier for low, intermediate and high voltage bias regimes (Simmons, J. G., *Journal of Applied Physics* 34, 1793-1803 (1963)). We had wrongly reported that the work function of the electrodes had been fixed at 5 eV during the fitting procedure. In fact, we only fixed the junction area at 100 nm^2 and fitted for both barrier height and gap width. However, we find that the 1D transport model developed by Mangin et al. (PHYSICAL REVIEW B 80, 235432 2009), which was suggested to us by Reviewer 3 (comment #8 and #9) is better suited for the purpose of estimating gap width, since it does not require choosing a seemingly arbitrary value for the cross-sectional area. The gap widths we obtain for our devices range from 0.8 to 1.6 nm, with one outlier of 2.9 nm. We find that the barrier heights resulting from fitting the model to our data are as expected significantly lower than the work function of clean Au in vacuum, and range from 1.2 to 3.5 eV, with an average of 2.6 eV (N=17 devices). All the fit parameters and gap widths reported now were derived using this 1D model: the fit curves for five devices are shown in the revised SI (page 11–20) and fit parameters for all bare tunneling junctions to which fits were performed are shown in the Supplementary Table 1 (SI page 21). We have also significantly modified a paragraph in the revised manuscript to report the correct values of gap width, the model and fitting procedure used to obtain them, and to clarify the description of the experiment already reported in that paragraph (page 4, line 103–127).

Reviewer #3 (Remarks to the Author):

Reviewer comment:

Report about manuscript: "Massively parallel fabrication of gold break junctions featuring sub-3 nm gaps " by Dubois et al.

The authors report a fabrication method for a large number of metallic nanogaps with yield of around 8% and relatively high integration density. The work is interesting and might be published, but I have several remarks.

Our response:

We thank the reviewer for the careful analysis of our work, and for the several remarks which have helped us to substantially improve the presentation and content of our work.

Reviewer comment:

1. I find the term break junction misleading. It is true that the nanogaps have been fabricated by breaking a formerly continuous wire, but this is not important.

Our response:

As correctly pointed out by the reviewer, the previous version of our manuscript did not explicitly mention the way in which crack-defined break junctions (CDBJ) described in our study differed from mechanically controllable break junctions (MCBJ): that the nanogap in CDBJ is fixed at the time of fabrication and is not tunable thereafter.

We have now changed the title to include the term "crack defined" to make it clear to the reader that the break junctions in our study were not fabricated using other techniques commonly associated with break junctions, namely MCBJ or STM-BJ. We have added the phrase "with fixed gap width" to a sentence in the abstract to again make the lack of gap-tunability of CDBJ evident. We have also clearly pointed out this distinction in the Introduction (page 1, line 7–9) and the Discussion (page 5–6, line 188–190) sections of the revised manuscript, so that the reader immediately understands this limitation of CDBJ.

Reviewer comment:

2. The term break junction implies that the junction is formed while breaking and is tunable and is not applied to prebroken nanogaps. As the authors state correctly electromigrated nanogaps are called breakdown junctions and not break junctions.

Our response:

We thank the reviewer for raising this issue regarding the proper terminology to use to describe our work. We have since debated the choice between the terms break junction and breakdown junction for our crack-defined nanogaps: Break junction was an appropriate choice considering the nature of formation of the nanogaps, namely the mechanical breaking of a previously unbroken constriction; breakdown junction was appropriate in terms of the similar feature of nanogaps formed by electromigrated breakdown to ours, namely the absence of gap tunability after fabrication.

We decided to keep the label “break junction” because the term breakdown is strongly associated with an electrical breakdown process being involved in the nanogap fabrication. However, to avoid potential misunderstanding by the readers, we have now explicitly stated in the revised manuscript– in the abstract and in the discussion section, that the CDBJ described in our study have a gap width that is fixed at the time of fabrication and cannot be changed later, unlike MCBJ and STM-BJ. We have also changed the title to include the term crack-defined. We believe that with these revisions we have made to the manuscript, in the text (page 1, line 7–9; page 5–6, line 188–190), to the abstract and to the title (with the addition of the classifier “crack-defined”), to address this and the previous comment, we will avoid any confusion regarding the capability of CDBJ, in comparison to MCBJ and STM-BJ.

Reviewer comment:

3. There is one important variant of break junctions which is not mentioned at all, namely STM break junctions. They are at least as common for contacting molecules as MCBJs and EBDJs. They are correctly named break junctions because of the in situ formation of the molecular junction.

Our response:

We thank the reviewer for bringing this omission to our notice. We have now mentioned STM-BJ in the revised manuscript and also pointed out their similarity with MCBJ, and distinction from CDBJ, including two new relevant references (page 1, line 5–9).

Reviewer comment:

4. Besides the tunability the in situ formation of the molecular junctions is important because freshly broken and therefore clean metal surfaces are formed upon each breaking. Fresh metal surfaces are important for the formation of molecular junctions.

Our response:

Following this suggestion and the following suggestion, as well as suggestions of Reviewer 2, we have carried out new experiments to demonstrate the formation of molecular junctions using our CDBJ. The molecule we use is oligo(phenylene ethynylene) (OPE3) with acetyl protected thiol groups (OPE3-SAc). OPE3 is a widely studied conjugated reference molecule. Of 13 new devices identified to be tunneling before OPE3 deposition, we formed molecular junctions in six of them. The formation of molecular junctions is indicated by a large increase in low bias conductance of the junctions after molecule deposition: in 5 of 6 cases this increase in conductance was more than one order of magnitude. Moreover, the logarithmic conductance values group around $2.4 \times 10^{-4}G_0$ after deposition, which is close to the values of $1-3 \times 10^{-4}G_0$ found in MCBJ measurements on OPE3-SAc. OPE3-SAc was deposited on the CDBJ using a wet process, starting with the cleaning of our junctions by immersion in dichloromethane. This or the original state of the junction itself seems to be adequate to leave free sites on the Au electrodes for molecular junctions to form.

We have added a new section titled “Demonstration of molecular junctions formed using CDBJ” to the revised manuscript where these experiments are described (page 5, line 165–184). A section in the methods section of the revised manuscript titled “Fabrication and characterization of molecular junctions” described the process of molecular deposition and the related electrical characterization (page 8, line 299–307). Figure 4 in the revised manuscript presents a schematic of the molecular junction and a I-V histogram of one molecular junction. Supplementary Figure 8 presents the I-V histograms under ambient conditions for all six molecular junctions, alongside the

I-V curves before molecule deposition (SI page 23–24). We have also added the phrase “for molecular devices” to the title to reflect the fact that CDBJ could be used to form molecular junctions.

Reviewer comment:

5. The authors claim that their method is suitable for forming molecular junctions but they do not give any evidence for this claim. He show that with a certain yield they can form tunnel contacts. The spread of the tunnel width is indicated wit 1.2 to 3 nm-. This spread is much too much to form single-molecule junctions with reasonable distribution of conductance. The authors must at least give one demonstration of a successful formation of a molecular junction.

Our response:

As we have responded to the reviewer’s suggestion to demonstrate the formation of molecular junctions using CDBJ in the previous remark. Here we will address the topic of gap width and its implications in response to this remark. In our responses to remarks 7, 8 and 9 below, we will describe that we have now fitted tunneling I-V characteristics to a 1D transport model to determine the gap width of our junctions assuming a symmetric potential barrier, as suggested by the reviewer. The gap widths so obtained range from 0.8 to 2.9 nm. However, barring one outlier at 2.9 nm, the next highest value is 1.6 nm, and the average gap width excluding the outlier is 1.3 nm. Therefore, the spread in gap widths (estimated from fits to the 1D model) is much narrower than we previously thought. The fit parameters for all devices used to substantiate this claim are provided in Supplementary Table 1 (SI page 21) and the 1D model and fitting procedure is described in Supplementary Note 1 (SI page 9).

The gap widths of the six junctions which formed molecular junctions range from 0.7-1.1 nm before molecule deposition. As the reviewer correctly predicted, the conductance of the molecular junctions is however distributed over a much larger range– 3.6×10^{-5} to $2.7 \times 10^{-2}G_0$, with the logarithmic conductance values grouping around $2.4 \times 10^{-4}G_0$. In addition to the gap width, we believe that different junction configurations which lead to variations in tunnel coupling strength between the molecules and Au electrodes to cause this wide range of molecular conductance. Since an investigation into the cause of this spread is beyond the scope of this study, we have not commented on this point in the manuscript.

Reviewer comment:

6. The authors claim to see Fowler Nordheim tunneling. From the data given in the SI it is not clear whether the tunneling is Fowler-Nordheim type. In order to support the claim they should check their data in a Fowler-Nordheim representation. To come into the Fowler-Nordheim regime the applied voltage must exceed the tunnel barrier height. The authors indicateto have used the bulk value for surfaces in vacuum (~ 5 eV). The applied voltage in the cases shown is always below 5 V. Thus, they cannot prove whether it is Fowler-Nordheim or not.

Our response:

We thank the reviewer for this comment. It is correct that the data we presented previously did not provide direct evidence for the observation of FN tunneling. We have now modified the figures in the SI to also show the FN representation of our I-V data (SI page 11–20).

Our claim regarding the observation of FN tunneling was based on the observation that in most cases $\ln(I/V^2)$ is increasing for decreasing values of $(1/V)$ for a significant portion of the applied bias in our measurements. However, we have since realized that this is not adequate proof to demonstrate FN tunneling, since the change from decreasing to increasing $\ln(I/V^2)$ can occur at bias values lower than the actual barrier height even in an ideal data set generated by Simmon's model for a symmetric barrier.

Because of this, and also because the observation of FN tunneling in a bare tunneling gap is not important to establish whether a crack-defined break junction is tunneling or not, which is what we want to demonstrate in our study, we have now removed all references to FN tunneling from the revised manuscript (page 3, line 73; page 4, line 120; page 5, line 157).

Reviewer comment:

7. The authors say to have used the Simmons model there are so many Simmons models around, the authors should indicate which approximation, which assumption for the tunnel barrier they used.

Our response:

We thank the reviewer for bringing this crucial omission to our attention. In this revised version of our work, we have used a 1D transport model across a trapezoidal barrier, which allows different barrier heights at the left and right electrodes. This model was developed by Mangin et al. (PHYSICAL REVIEW B 80, 235432 2009) and suggested to us by the reviewer. However, we got better fits for the I-V characteristics presented in the previous version of our SI by fitting to a symmetric barrier (with equal barrier heights at the left and right electrodes). We have updated the text to clearly mention the model we used in the revised manuscript (page 4, line 121–124) and added Supplementary Note 1 with the model description, fitting procedure and the reasoning for our choice of a symmetric barrier (SI page 9).

In our initial submission we used the 3D form of Simmon's model for a symmetric rectangular barrier, using the approximate analytical expressions reported in Simmon's original paper for tunneling current densities for low, intermediate and high voltage bias regimes. We had wrongly reported that the work function of the electrodes had been fixed at 5 eV during the fitting procedure. In fact, we only fixed the junction area at 100 nm² and fitted for both barrier height and gap width. However, we find that the 1D transport model which was suggested to us is better suited for the purpose of estimating gap width, since it does not require choosing a seemingly arbitrary value for the cross-sectional area. All the fit parameters and gap widths reported now are derived using this 1D model: the fit curves for five devices are shown in the SI (page 11–20) and fit parameters for all bare tunneling junctions to which fits were performed are shown in the Supplementary Table 1 (SI page 21).

Reviewer comment:

8. The gap width has been determined from the Simmons model using the value ~ 5 eV for the work function. This is highly questionable for ligament –formed structures that have been exposed to atmosphere, as is shown e.g. PHYSICAL REVIEW B 80, 235432 2009 written by the authors of their reference 25.

Our response:

We thank the reviewer for this comment, which is absolutely correct. As we have mentioned in response to the previous remark, our statement regarding the assumption of 5 eV for the electrode work function in the manuscript was an error in writing. As also mentioned in our response to the previous remark, we have changed the model used to fit the tunneling I-V characteristics, and now use a 1D model. We find that the barrier heights resulting from fitting the model to our data are significantly lower than the work function of clean Au in vacuum, and range from 1.2 to 3.5 eV, with an average of 2.6 eV (N=17 devices). This is however significantly higher than the average of 0.7 eV reported by Mangin et al. (PHYSICAL REVIEW B 80, 235432 2009). The gap width and work functions obtained as fit parameters to the 1D model are tabulated in Supplementary Table 1 and discussed in the SI (SI page 21).

Reviewer comment:

9. The Simmons model (in the usual variation for a rectangular barrier) contain three fitting parameters, which are not independent of each other: the work function the tunnel gap and the cross section. Giving the limited voltage window I assume that there are several combinations of these three parameters which give rise to fits of comparable quality. A basic quality check is to use also the opposite polarity of the voltage bias and verify the symmetry. In the reference given above a decent analysis of tunnel data with a one-dimensional model is given. This model contains for symmetric electrodes (symmetric IVs) only two parameters. It is worth testing this model.

Our response:

We thank the reviewer for this comment. As mentioned previously, we have implemented the 1D model reported in the work of Mangin et al. (PHYSICAL REVIEW B 80, 235432 2009) suggested by the reviewer and tested our tunneling I-V characteristics by fitting to a symmetric barrier and presented our findings in the Supplementary Information in Supplementary Note 1 and Supplementary Table 1.

We have performed new measurements to identify tunneling junctions to form molecular bridges with. In these measurements we restricted the voltage bias levels to below 0.5 V and collected measurements for positive and negative bias levels (the old measurements were performed with positive bias only). We find that these tunneling junctions display highly symmetric I-V characteristics, as can be seen from the six junctions for which these plots have been presented in Supplementary Figure 8 (SI page 23–24, middle column).

Reviewer comment:

10. I wonder whether the nanogap structures keep their geometry during SEM inspection. To achieve the high spatial resolution the electron density of the imaging electron beam is usually relatively high. From own experience I know that motion of atoms takes play even on the several nanometer scale. It might be that because of the relatively low acceleration voltage used here the bombardment is less invasive than with usual SEMs, but the authors should give evidence for the noninvasiveness of their characterization method. One possibility would be to vary the imaging current and check the stability of the images.

Our response:

We thank the reviewer for the opportunity to further expand and clarify on this point. In our study, we have used SEM imaging primarily as a means to correlate a device classified as “ohmic”,

“tunneling” or “isolated” based on electrical measurements with its morphology. As already described in the manuscript (page 4, line 103–143), in 86 devices we first performed electrical measurements and then collected SEM images. Based on the correlation we observed between junction morphology and “electrical” classification in these devices, we sought to reverse the sequence, i.e. perform SEM imaging before electrical characterization, and to determine whether the correlation was still strong. Therefore, in 180 devices, we first performed SEM imaging and identified 31 devices as likely to be tunneling, based on their appearance. Then we performed electrical measurements only on this subset of devices to identify those which were nanogaps (“tunneling”; 15/31), and those which were ligaments (“ohmic”; 16/31) as also already described in the manuscript (page 5, line 154–162). Since we realize that our previous formulation was unclear, we have modified the revised manuscript to clarify that in these junctions electrical characterization was done after SEM imaging was used to identify possible tunneling junctions (page 5, line 154–157; page 8, line 281–283).

As the reviewer correctly identified, we did not show evidence to prove that the SEM imaging, in this second set of devices in particular, is non-invasive. We have now collected a series of SEM images of our devices (three images each of six devices) under conditions we regularly use to obtain high resolution images (3.5 kV acceleration voltage, 4 mm working distance, 150 kx magnification). These images show that there is no noticeable reconfiguration of the tips due to SEM imaging. These images are now presented in the revised Supplementary Information (SI page 8, Supplementary Figure 7) and are referred to in the revised manuscript in the section on Morphological characterization (page 8, line 286–288).

Reviewer comment:

11. I wonder what the progress is compared to their reference 27. It is mentioned in the reference list but I do not see where it is quoted in the manuscript.

Our response:

We thank the reviewer for bringing this error to our attention. Reference 27 was a duplicate of Reference 21, this was the reason it was not cited in the manuscript. We have now removed the original Reference 27 from the revised manuscript.

Reviewer #1 (Remarks to the Author):

The authors have addressed my concerns (and those of the other referees) in their revisions. As part of the revision process they have introduced considerably more and revised material regarding the electrical characterization of the junctions, including demonstration of electrical measurements following exposure to OPE molecules. In general the manuscript is improved, and I continue to favor publication, though I do have a couple of minor points that I hope can be addressed.

1) With the introduction of the molecular junction measurements, the authors could be more clear in their description of the procedure. They say that they investigated 13 junctions in this part of the study, finding that 6 showed apparent molecular conduction after exposure to the oligomer solution. How were those 13 junctions selected? That is, was some larger ensemble of junctions measured pre-exposure, to identify junctions that had measurable tunneling characteristics? How large was that ensemble, and what were the criteria used to pick the 13 junctions in question?

2) Under what conditions were the room temperature electrical measurements made of both bare junctions and molecule-exposed junctions? When the authors say "ambient" conditions, does that mean in open air (including moisture, etc.)? Was a probing station used, or were leads wire-bonded to a chip carrier? (I ask partly out of interest regarding the static sensitivity and robustness of these devices.)

Reviewer #2 (Remarks to the Author):

The answers to the comments by the referees are extensive and satisfactory. The addition of data for molecular junctions adds to the value of the paper.

I advise acceptance of the manuscript for publication.

Reviewer #3 (Remarks to the Author):

The authors have very thoroughly revised the manuscript. The response shows that the original version of the submitted manuscript was not mature, since it contained contradictory and even erroneous statements. In that sense it would have been better to check the consistency of the

material before submission instead of letting the reviewers do this work. In the revised material, all criticism has been addressed, new data has been included and the reviewers' suggestions have been taken seriously. I have no further suggestion for improvements and recommend publication of the manuscript as it is now.

Reviewers' comments:

Reviewer #1 (Remarks to the Author):

Reviewer comment:

The authors have addressed my concerns (and those of the other referees) in their revisions. As part of the revision process they have introduced considerably more and revised material regarding the electrical characterization of the junctions, including demonstration of electrical measurements following exposure to OPE molecules. In general the manuscript is improved, and I continue to favor publication, though I do have a couple of minor points that I hope can be addressed.

Our response:

We thank the reviewer for recommending the publication of our manuscript.

1) With the introduction of the molecular junction measurements, the authors could be more clear in their description of the procedure. They say that they investigated 13 junctions in this part of the study, finding that 6 showed apparent molecular conduction after exposure to the oligomer solution. How were those 13 junctions selected? That is, was some larger ensemble of junctions measured pre-exposure, to identify junctions that had measurable tunneling characteristics? How large was that ensemble, and what were the criteria used to pick the 13 junctions in question?

Our response:

We thank the reviewer for these questions, which have made us realize that the presentation of the molecular measurements was unclear. Bare tunneling junctions for molecular junction experiments were identified by systematically performing I-V characterization using bipolar voltage sweeps up to 500 mV in magnitude on 540 devices on 6 cells on the wafer. 13 of the 540 devices displayed measurable tunneling currents at these voltage levels. Of these 13 junctions, 10 were from a subset of 180 devices with the most promising design parameters already discussed in the Results section of the manuscript. The remaining 3 junctions were from adjacent columns of devices in the swept cells whose design parameters were not optimal, thereby explaining the low yield of tunneling junctions observed for this subset of devices. We have now described this procedure in Methods in the section titled "Fabrication and characterization of molecular junctions".

Reviewer comment:

2) Under what conditions were the room temperature electrical measurements made of both bare junctions and molecule-exposed junctions? When the authors say "ambient" conditions, does that mean in open air (including moisture, etc.)? Was a probing station used, or were leads wire-bonded to a chip carrier? (I ask partly out of interest regarding the static sensitivity and robustness of these devices.)

Our response:

All room temperature measurements in this study were made in open atmosphere, that is, with no control of the device temperature or the ambient atmosphere it was exposed to. Therefore, the devices were exposed to moisture in the air. However, while performing low temperature measurements the lid of the cryogenic probe station was sealed and the chamber was evacuated to place the devices in vacuum.

All electrical measurements in our study, at all temperatures, were performed in a probe station. Wire bonding was not used in our study. We have now stated clearly in Methods in the section titled “Fabrication and characterization of molecular junctions” that room temperature measurements were performed in a probe station and in open atmosphere.

Reviewer #2 (Remarks to the Author):

Reviewer comment:

The answers to the comments by the referees are extensive and satisfactory. The addition of data for molecular junctions adds to the value of the paper. I advice acceptance of the manuscript for publication.

Our response:

We thank the reviewer for appreciating the changes we made in revising the manuscript and for recommending its acceptance for publication.

Reviewer #3 (Remarks to the Author):

Reviewer comment:

The authors have very thoroughly revised the manuscript. The respose shows that the original version of the submitted manuscript was not mature, since it contained contradictory and even erroneous statements. In that sense it would have been better to check the consistency of te material before submission instead of letting the reviewers do ths work. In the revised material, all critcism has been addressed, new data has been included and the reviewers' suggestions have been taken seriously. I have no further suggestion for improvements and recommend publication of the manuscript as it is now.

Our response:

We thank the reviewer for their favorable appraisal of our revisions and for supporting the publication of our manuscript.